

# Development of medical informatics in China over the past 30 years from a conference perspective and a Sino-American comparison

Jun Liang[1], Kunyan Wei[2], Qun Meng[3], Zhenying Chen[4], Jiajie Zhang[5] and Jianbo Lei[6,7]

[1] IT Center, Second Affiliated Hospital, School of Medicine, Zhejiang University, Hangzhou, Zhejiang Province, China
[2] Department of Gastroenterology, Affiliated Hospital of Southwest Medical University, Luzhou, Sichuan Province, China
[3] Center for Statistics and Information, National Health and Family Planning Commission of China, Beijing, China
[4] Library of Zhejiang University, Hangzhou, Zhejiang Province, China
[5] School of Biomedical Informatics, University of Texas Health Sciences Center, Houston, TX, United States of America
[6] Center for Medical Informatics, Peking University, Beijing, China
[7] School of Medical Informatics and Engineering, Southwest Medical University, Luzhou, Sichuan Province, China

Corresponding author
Jianbo Lei, jblei@hsc.pku.edu.cn

## ABSTRACT

**Background**. As the world's second-largest economy, China has launched health reforms for the second time and invested significant funding in medical informatics (MI) since 2010; however, few studies have been conducted on the outcomes of this ambitious cause.

**Objective**. This study analyzed the features of major MI meetings held in China and compared them with similar MI conferences in the United States, aiming at informing researchers on the outcomes of MI in China and the US from the professional conference perspective and encouraging greater international cooperation for the advancement of the field of medical informatics in China and, ultimately, the promotion of China's health reform.

**Methods**. Qualitative and quantitative analyses of four MI meetings in China (i.e., CMIAAS, CHINC, CHITEC, and CPMI) and two in the US (i.e., AMIA and HIMSS) were conducted. Furthermore, the size, constituent parts and regional allocation of participants, topics, and fields of research for each meeting were determined and compared.

**Results**. From 1985 to 2016, approximately 45,000 individuals attended the CMIAAS and CPMI (academic), CHINC and CHITEC (industry), resulting in 5,085 documented articles. In contrast, in 2015, 38,000 and 3,700 individuals, respectively, attended the American HIMSS (industry) and AMIA (academic) conferences and published 1,926 papers in the latter. Compared to those of HIMSS in 2015, the meeting duration of Chinese industry CHITEC was 3 vs. 5 days, the number of vendors was 100 vs. 1,500+, the number of sub-forums was 10 vs. 250; while compared to those of AMIA, the meeting duration of Chinese CMIAAS was 2 vs. 8 days, the number of vendors was 5 vs. 65+, the number of sub-forums was 4 vs. 26. HIMSS and AMIA were more

open, international, and comprehensive in comparison to the aforementioned Chinese conferences.

**Conclusions**. The current MI in China can be characterized as "hot in industry application, and cold in academic research." Taking into consideration the economic scale together with the huge investment in MI, conference yield and attendee diversity are still low in China. This study demonstrates an urgent necessity to elevate the medical informatics discipline in China and to expand research fields in order to maintain pace with the development of medical informatics in the US and other countries.

## INTRODUCTION

### Development of the discipline of medical informatics

Medical informatics (MI) is a multidisciplinary field in which researchers pursue scientific exploration, problem-solving, and decision-making to facilitate the effective use of biomedical data, information, and knowledge for the improvement of human health (*Kulikowski et al., 2012*). First proposed in 1970 (*Haux, 2010*), medical informatics currently includes 20 subfields, such as clinical informatics (*Kulikowski, 2007*; *Sehuemie, Talmon & Moorlna, 2009*); over time, the focus of MI research has shifted from hardware and software innovation to software systems and information-processing models, particularly data and knowledge description and management, computerization, and evaluation (*Kulikowski, 2007*). Medical informatics has been recognized worldwide as an emerging, independent, and important interdisciplinary field.

### High-level Design and Industry Development of Hospital Informatics in China

Rapid economic development and recent health care reforms in China have greatly facilitated the development of hospital informatics. In 2010, as China became the world's second largest economy, the Chinese government initiated its second health care reform, which is the largest since the founding of the P.R. China, with an announcement that "medical information systems and population health informatics" would be one of the "four pillars, eight posts" supporting reforms in the health care field (*Qun, 2011*). Guided and stimulated by these policies, the Chinese government and healthcare informatics users have made substantial investments in population health informatics and hospital informatics. During the past and 12th 5-year plan, encompassing the period between 2011 and 2015, the Chinese government began investing a considerable amount of money for the purpose of promoting the construction of health information. The central government alone directly appropriated 1.5 billion US dollars for village-level and country-level information construction projects in the midwest region's 22 provinces. The former included health information system construction for primary health care institutions covering all the public township and community health institutions; in addition, the central
government invested 45 million US dollars for the health information projects of 16 health care reform pilot cities, 120 million US dollars for 16 provincial information platform construction projects, and 60 million US dollars for telemedicine information system projects covering 22 provinces. By 2020, the government will further invest more than 3 billion US dollars for the "national electronic health information systems engineering" project (*Hu, Xin & Qun, 2016*). In 2015, the market of hospital informatics alone reached 5 billion US dollars; by 2020, this market is projected to exceed 14.5 billion US dollars, at an annual compound growth rate of ≥24% (*Luo, 2014*).

Given the space limitations of this paper, readers may refer to Appendix A-14 (*Evolution of Hospital Informatics in China*) if interested in additional information.

## Development of the medical informatics discipline in China is relatively behind that of the US

In China, the discipline of medical informatics is primarily characterized by a mismatch between the developmental model and hospital informatics. Medical informatics had a late start, with roots in traditional medical information and library science. Originating in China during the 1970s, this discipline has been evolving for over three decades (*Li & Liu, 2014*). Because various biomedical and computer technologies are widely used in the medical field in China, the research focus of medical informatics has expanded from library informatics and information technology at information organizations, to information theory research, business systems, and the construction and application of infrastructure in the medical field. At present, medical informatics research in China focuses on Health Information Technology (HIT)-related organizations and includes HIT application and evaluation (e.g., hospital information systems); medical information research and services; and medical information resource development, retrieval, and services. Hospital information systems are primarily implemented and applied by each hospital; medical information research and services are the major obligation of medical information research institutes; and medical information resources development, retrieval, and services are the major obligation of medical libraries.

An evolving interdisciplinary subject in China, medical informatics (MI) remains relatively backward; until recently, the concept was not clearly defined or widely accepted (*Lei et al., 2016*). As with any discipline, MI requires systematic support—inclusive of related mainstream journals, publications, agencies, associations, and academic institutions—in order to become established and advance (*Dong, 2004*). However, as mentioned previously, medical informatics in China is subject to the influence of traditional medical information and library science, as well as historical and other challenges. Nevertheless, with rapid economic growth, the importance that the government ostensibly places on the development of medical informatics—in the form of government-led top design and huge investments—and China's myriad medical institutions and its massive population, there is now an unprecedented opportunity for medical informatics to successfully evolve. To this end, it will be important to strengthen international cooperation so that China and other countries may learn from one another and promote the development of medical informatics.

**Table 1 Basic information on the CMIAAS, CHINC, CHITEC, and CPMI conferences.** CMIAAS, only 12 conferences have been held since 1981–2015; CPMI, 24 conferences have been held since 1993–2016; CHITEC, only 13 conferences have been held since 2004–2016; CHINC, 20 conferences have been held since 1997–2016.

| | Organizer | Inception | Schedule | Number of conferences held | Conference duration | Type | Scale of last Conference (number of attendees) |
|---|---|---|---|---|---|---|---|
| CMIAAS | China Medical Informatics Association | 1981 | Every 3 years | 12 | 1 day | Academic conference | 200+ (CMIAAS 2015) |
| CPMI | Medical Informatics Branch, Chinese Medical Association | 1993 | Annual | 24 | 1 day | Academic conference | 500+ (CPMI 2016) |
| CHITEC | China Institutes of Health Information | 2004 | Annual | 13 | 2 days | Industry conference | 5,000+ (CHITEC 2016) |
| CHINC | Committee on Information Management, Chinese Hospital Association | 1997 | Annual | 20 | 3 days | Industry conference | 3,500+ (CHINC 2016) |

[1]Including annual symposium and joint summits, for the sake of convenience, together as AMIA.

This paper focuses on four national mainstream medical informatics conferences and compares their data with the data from two international medical informatics conferences based in the US (i.e., the AMIA [1] and HIMSS meetings). This aims to explore the differences and identify the lessons learned in order to aid Chinese researchers and their international colleagues in understanding the characteristics of medical informatics in China and how progress might be made. Here, the overall goal is sharing China's experience with other countries, for the purpose of promoting the exchange of knowledge worldwide for the improvement of China's medical informatics as well as that of other nations if possible that are likely to lag in the discipline.

## BACKGROUND

### Mainstream Chinese medical informatics conferences and their evolution

There are four mainstream Chinese medical informatics conferences: the China Hospital Information Network Conference (CHINC), the Chinese Medicine Information Association Annual Symposium (CMIAAS), the China Proceedings of Medical Informatics (CPMI) and the China Health Information Technology Exchange Conference (CHITEC). Basic information on these four conferences is presented in Table 1 below.

Please refer to Appendix A-2 (*Evolution of Chinese Mainstream Medical Informatics Conferences*) for more information regarding these conferences, including descriptions of these conferences, related organizations and their URLs.

## MATERIALS AND METHODS

### Selection of mainstream MI conference in China

According to the PRISMA (Preferred Reporting Items for Systematic Reviews and Meta-Analysis) method (*Moher et al., 2009*), the selection criteria are defined as follows:

1. The academic topics of the meeting must be related to medical informatics, including but not limited to: clinical informatics, bioinformatics, drug informatics, nursing informatics, public health informatics, medical image informatics, etc.;

2. The meeting must be organized by a national academic organization or international academic organization in China;
3. The meeting must call for papers through academic journals or an international open approach;
4. The meeting must be continuous and have been held at least five times.
5. The meeting must have a certain degree of visibility, academic credibility and influence in China's medical informatics community.

Ultimately, we chose four HIT meetings as the major meetings in mainland China, namely: CMIAAS hosted by CMIA (the only country representative of IMIA), CPMI hosted by CSMI, CHINC hosted by CHIMA and CHITEC hosted by CHIA.

## Collection of data

China's top three biggest literature databases were searched: the VIP database of Chinese scientific and technical journals, the Chinese National Knowledge Infrastructure, and the Wanfang data retrieval platform. These databases were searched for "会议论文集 (meeting proceedings)." Keywords included the Chinese name of the conference (中国医药信息学大会 [CMIAAS], 中华医院网络信息大会 [CHINC], 中国卫生信息技术交流大会 [CHITEC], and 中华医学会全国医学信息学术会议 [CPMI]), the Chinese organizer of the conference (中国医药信息学会 [China Medical Informatics Association] [CMIA], 中国医院协会信息管理专业委员会 [the Committee on Information Management, Chinese Hospital Association] [CHIMA], 中国卫生信息学会 [China Institutes of Health Information] [CHIA], 中华医学会医学信息分会 and [the Medical Informatics Branch, Chinese Medical Society] also known as [Chinese Society of Medical Information] [SCMI]), as well as the English acronym of each meeting (CMIAAS, CHITEC, CHINC, and CPMI). Meeting proceedings between 1985 and 2015 were retrieved. Furthermore, related supporting information, together with biographical references, was also searched, such as the purpose, business scope, and "call for papers" about each meeting.

Data about the Healthcare Information and Management Systems Society (HIMSS) and the American Medical Informatics Association (AMIA), which are the two major US medical informatics meetings, were obtained from literature reviews (*Fickenscher, 2013a*; *Fickenscher, 2012a*; *Fickenscher, 2012b*; *Fickenscher, 2013b*; *Fickenscher, 2013c*; *Fickenscher, 2013d*; *Middleton, 2014*; *Ravvaz et al., 2015*; *Shortliffe, 2011a*; *Shortliffe, 2011b*) as well as the "letter of welcome" for AMIA and HIMSS and websites about meeting organizers.

## Extraction of data

Information about target meetings from the literature and databases was exported, for instance:
1. Meeting size, including the meeting's name, who held it and when it was held, its schedule, the duration, the number of participants, etc.
2. General information regarding the proceedings, for instance, title, author affiliation, publication year, meeting name, and where the author lives (a self-developed tool is extracted).
3. Subjects of conference proceedings were gathered and analyzed, in addition to the "call for papers" of the CMIAAS, CPMI, CHINC, and CHITEC (2000–2015) and the
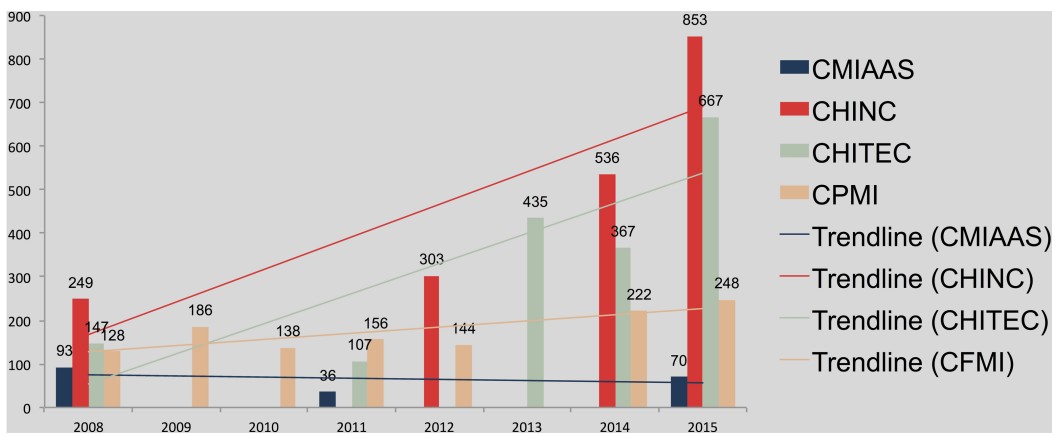

**Figure 1** Conference proceedings of the CMIAAS (2008–2015), CHINC (2008, 2012, 2014, 2015), CHITEC (2008, 2011, 2013–2015), and CPMI (2008–2012, 2014, 2015).

organizer's business scope. According to the results, these meetings mainly dealt with highly similar topics in 10 fields of research, including but not limited to hospital informatics, regional as well as grassroots health informatics and telemedicine.

## Data analysis

EndNote X7, EXCEL 2011, and Python were used for a preliminary analysis of the general information assembled, along with proceedings topics. The results of the analysis were described from the perspective of the actual developmental circumstances of medical informatics and HIT in China.

## RESULTS

### Overall volume and trend of conference proceedings

We searched three mainstream Chinese databases and retrieved a total of 6,681 papers from the proceedings of four conferences (1985 to 2015). For the purposes of this study, we intentionally selected conference proceedings published after 2008 because most conference papers (76% 5,085 papers) were published on or after 2008, when publication and database entries for medical informatics conference proceedings became more standardized and comprehensive in China. This choice contributed to the alignment and data analysis of the proceedings. Moreover, during that time, the Chinese government was preparing to propose its "second health care reform," and HIT application was about to burst onto the scene.

Figure 1 shows Chinese MI conference proceedings trends, by conference, from 2008–2015, excluding some unavoidable gaps. It can be seen clearly that the growth rate of industry conference submission is much higher than that of academic conferences, and CMIAAS even had negative growth The overall trend of papers published in various Chinese MI conferences demonstrated a tendency of "hot in industry application, and cold in academic research". In addition, We observed that the CPMI proceedings were incomplete (i.e., the CPMI-2013 was missing) and, for CHINC and CHITEC, a large

percentage of the proceedings were missing, including the CHINC 2009–2011 and 2013, as well as CHITEC 2009, 2010, and 2012. We conducted preliminary descriptive statistical data analysis for each conference, with a focus on the continuity of proceedings entries in the literature databases. Note that, in PubMed, all AMIA proceedings since 1977 (annual symposium) and 2008 (joint summits) are available for download free-of-charge; in China, however, databases for the four mainstream medical informatics conferences are neither continuous nor unified, and downloading is not complimentary. Literature articles were distributed among three different databases, and some proceedings were downloadable only from the conference's association website.

In addition, we examined conference and editorial reviews of the proceedings and observed that, unlike international mainstream medical informatics conferences (e.g., AMIA), which implement a rigorous review process for submitted manuscripts (*Kulikowski, 2007*), Chinese medical informatics conferences have yet to implement a process of peer-review for submitted papers in most cases, only style and formatting requirements checking had been done; For some MI conferences in China, the acceptance rate was higher than 90% (*Li, 2015*; *Ya-min, Yan & Xiao-tao, 2011*; *Yajie et al., 2015*; *Yajie et al., 2014*). The main reason for such few submission and high acceptance is likely due to a weak foundation for medical informatics and an insufficient pool of qualified peer-reviewers.

## Number of attendees and trends

To compare the number of submitted conference papers and align the data across different conferences, we collected relevant information on all conference sessions from each conference website, as well as corresponding association websites. Note that CHITEC'16, CHINC'16, and CPMI'16 had finished, but the organizers have released on their homepages only attendance information such as number of participants, while the proceedings are not available in these databases. For CHITEC'16, CHINC'16, and CPMI'16, we have only the number of participants in the comparisons. Figure 2 depicts the trend of conference attendees from 2008–2016, excluding CMIAAS (2008–2015), showing that the number of attendees declined slightly at CMIAAS and increased by only 140 from 2008–2016 at CPMI. By contrast, the number of attendees increased dramatically at CHINC and CHITEC, by 3.9-fold and 8.3-fold, respectively, over 2008–2016, which is consistent with results described in this section. It is within expectation that the number of attendees was highly consistent with the trend of published proceedings, further confirmed the characteristic of Chinese MI development, "hot in industry application, and cold in academic research".

## Source of conference proceedings: types of first-author affiliations

Author affiliations were selected according to the requirements delineated in each conference's "call for papers," inclusive of medical institutions, universities, research institutions, and manufacturers. We referenced similar international evaluations and focused on first-author affiliations, which proved to be a good indicator of each type of

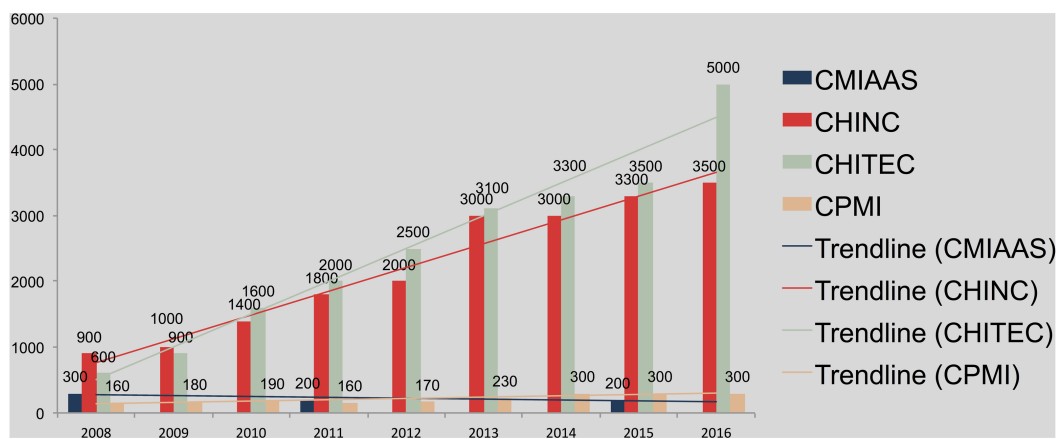

**Figure 2  The number of CMIAAS (2008–2015), CHINC (2008–2016), CHITEC (2008–2016), and CPMI (2008–2016) conference attendees.**

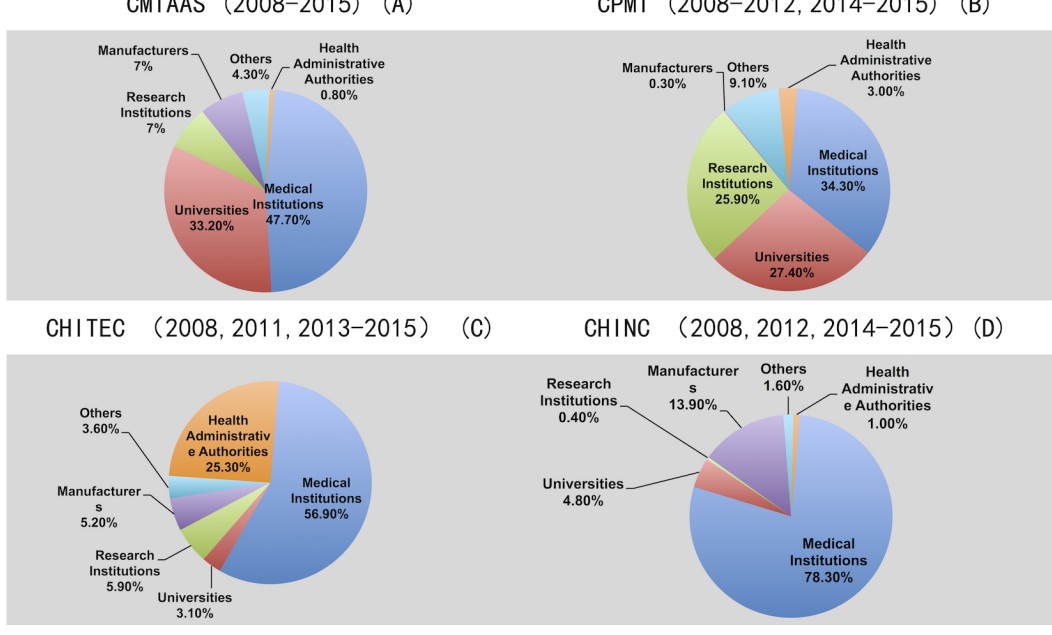

**Figure 3  Author affiliations at four mainstream Chinese medical informatics conferences (CMIAAS, CHINC, CHITEC, and CPMI).** Data show (A) author affiliations at CMIAAS, (B) author affiliations at CMPI, (C) author affiliations at CHITEC, (D) author affiliations at CHINC.

first-author affiliation. Figure 3 shows the type and percentage of first-author affiliations (% all affiliations) among available conference proceedings from 2008–2015.

Figure 3 shows a clear trend of author affiliations for four Chinese mainstream medical informatics conferences:

1. Most attendees (indirectly inferred from the distribution of authors because of data availability) of Chinese medical informatics conferences (CMIAAS, CHINC, CHITEC, and CPMI) were from "medical institutions," which differed from the attendees of annual HIMSS and AMIA conferences; most AMIA conferences attendees were from universities and research institutes, and the largest HIMSS attendee group was software providers in the HIT industry. Moreover, significantly more attendees were from "medical institutions" than from other institutions, whereas 78.3% of attendees at the CHINC were from medical institutions, which may reflect the purpose of the organizer, China Hospital Information Management Association (CHIMA), which focuses on domestic and foreign hospital information management. At the CMIAAS, 33.2% of attendees were from universities, which is consistent with MEDINFO attendees (*Kulikowski, 2014*). CHITEC is a semi-official mainstream Chinese medical informatics conference; thus, many attendees (>25%) were from "health administrative authorities." At the CPMI, attendees were evenly distributed among the various entities; such representation was most likely because the organizer SCMI was initially focused on traditional medical information and library science and information services—therefore, attendees represented universities, research institutes, or medical institutions. Unlike HIMSS, few attendees representing manufacturers submitted first-author papers to Chinese medical informatics conferences. At CHINC, 13.9% of attendees were manufacturers, whereas at the other three conferences, the percentage was ≤7%. Here, we use the distribution of the first author of the conference paper to simulate the distribution of the actual participants, because the accurate data of the actual participants can only be obtained from the various organizers and cannot be obtained from the public channels.

2. The AMIA and HIMSS conferences represent two different medical informatics exchange platforms: academic research (AMIA) and HIT application (HIMSS). Among mainstream Chinese medical informatics conferences, CHINC and CHITEC are similar to HIMSS and its focus on HIT applications. Most CHINC and CHITEC attendees were from medical institutions or administrative authorities, and the scale of these conferences was also large; in 2015, 667 papers were included in CHITEC proceedings and 853 in CHINC proceedings. Similar to AMIA, the CMIAAS and CPMI are medical informatics conferences that focus on academic exchange, as well as some HIT applications; most attendees were from medical institutions, and some attendees were from universities and research institutes. Their scale was much smaller; in 2015, only 70 papers were included in CMIAAS proceedings and 248 in CPMI proceedings. We compared the scale and attendee affiliations for these two types of conferences and observed that, even at academic conferences, the percentage of attendees from research institutes remained relatively low and that most papers focused on information systems application in medical institutions. This information provided indirect evidence for the postulation that "HIT is popular in industry application but unpopular in academic research in medical informatics" (*Lei et al., 2016*).
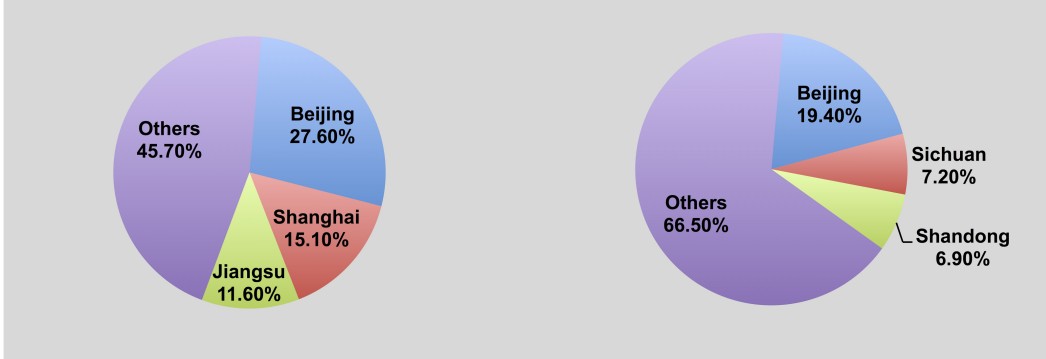

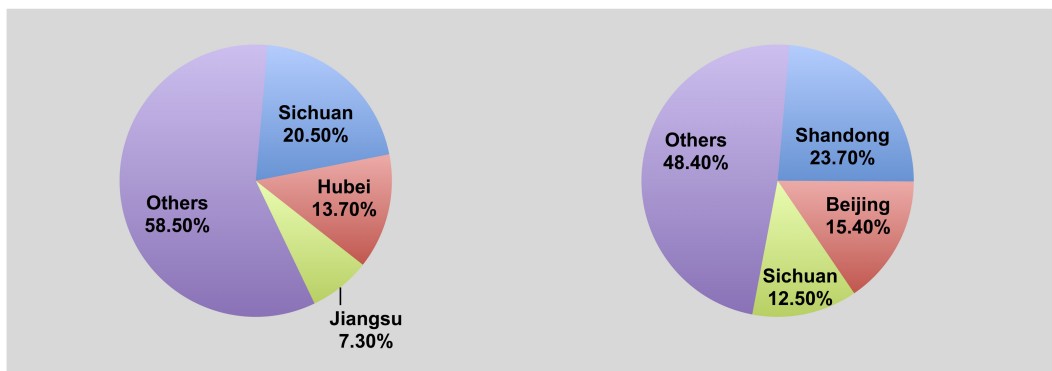

**Figure 4** **Geographic distribution of authors among the CMIAAS, CHINC, CHITEC, and CPMI conferences (the top three provinces and/or municipalities are listed).** Data show (A) geographic distribution of authors at CMIAAS, (B) geographic distribution of authors at CMPI, (C) geographic distribution of authors at CHITEC, (D) geographic distribution of authors at CHINC. (The top three provinces and/or municipalities are listed).

## Geographic distribution of conference proceedings first authors

We also analyzed the geographic distribution (e.g., municipality, province, etc.) of practitioners and researchers attending the previously described conferences from 2008–2015. Figure 4 shows the distribution of first authors (percentage, by geographic area) for each of the conference proceedings. The top three provinces and/or municipalities were selected, as shown below:

The academic disciplines and industry HIT development in different provinces and cities are unbalanced, showing the characteristics of "strong coastal areas in the East and weak inland areas in the West". This is because that comprehensive giant medical institutions, high-level medical information systems providers, research institutions are concentrated in a small number of eastern densely populated, medical and educational resources rich, economically developed municipalities and provinces.

We further compared and analyzed the statistics in Table 1 (above), and concluded the following:

1. Not surprisingly, Chinese medical informatics conferences were more regional and local than those of AMIA and HIMSS; that is, most papers were submitted by first authors from a small number of provinces and municipalities; other papers were submitted by authors outside Mainland China (as noted by the designation "Others"). Among the top contributors (provinces and municipalities) to these four conferences, Beijing and Sichuan were in three of the "top three" lists, and Jiangsu was in two of the "top three" lists. Because Beijing is the capital of China (*CHINA, 2014a*), this ranking was expected; Sichuan (*Shi & Zheng, 2008*) and Jiangsu (*CHINA, 2014b*) are economically developed areas with abundant medical, scientific, and technological resources and an active and mature HIT market. In addition, unlike international mainstream medical informatics conferences (e.g., AMIA and HIMSS) (*Maojo et al., 2012*), few practitioners or researchers outside Mainland China submitted papers to Chinese medical informatics conferences. Because the international influence of China's medical informatization is still very weak, the current positioning of these conferences is only as domestic conferences. Everything from the conference themes and service targets to the contents of publications reflects Chinese characteristics, and the conference exchange language is also Chinese.

2. Consistent with the economic development in different regions of China, the geographic distribution of attendees was also highly uneven. For all four conferences, more than one-third of attendees were from the top three provinces and/or municipalities (≥50% at CMIAAS and CHINC), indicating that, overall, less than half of the attendees were from the remaining 29 provinces, which is evidence of the uneven development of academic research and HIT applications in medical informatics throughout various regions of China.

## Distribution of conference proceedings topics

We browsed titles and abstracts of conference proceedings and classified papers for topic areas and statistically analyzed the results. We believe that the percentage of different topics covered in conference proceedings provides indirect evidence of current hot and cold topics in Chinese MI academic research and HIT application. The primary results (i.e., the distribution of topics for CMIAAS, CHINC, CHITEC, and CPMI conference proceedings [2008–2015]) are shown in Fig. 5.

Figure 5 shows the following characteristics related to the distribution of HIT applications and research areas among the four Chinese mainstream medical informatics conference proceedings:

1. "Health Information Technology (HIT) is popular in industry application but unpopular in academic research for medical informatics." Specifically, more than 50% of topics for the four conference proceedings were related to HIT applications in medical institutions and population health informatics, such as "hospital informatics" and "public and regional, and grassroots health informatics as well as telemedicine." These topics are consistent with the "four pillars, eight posts" policies of current
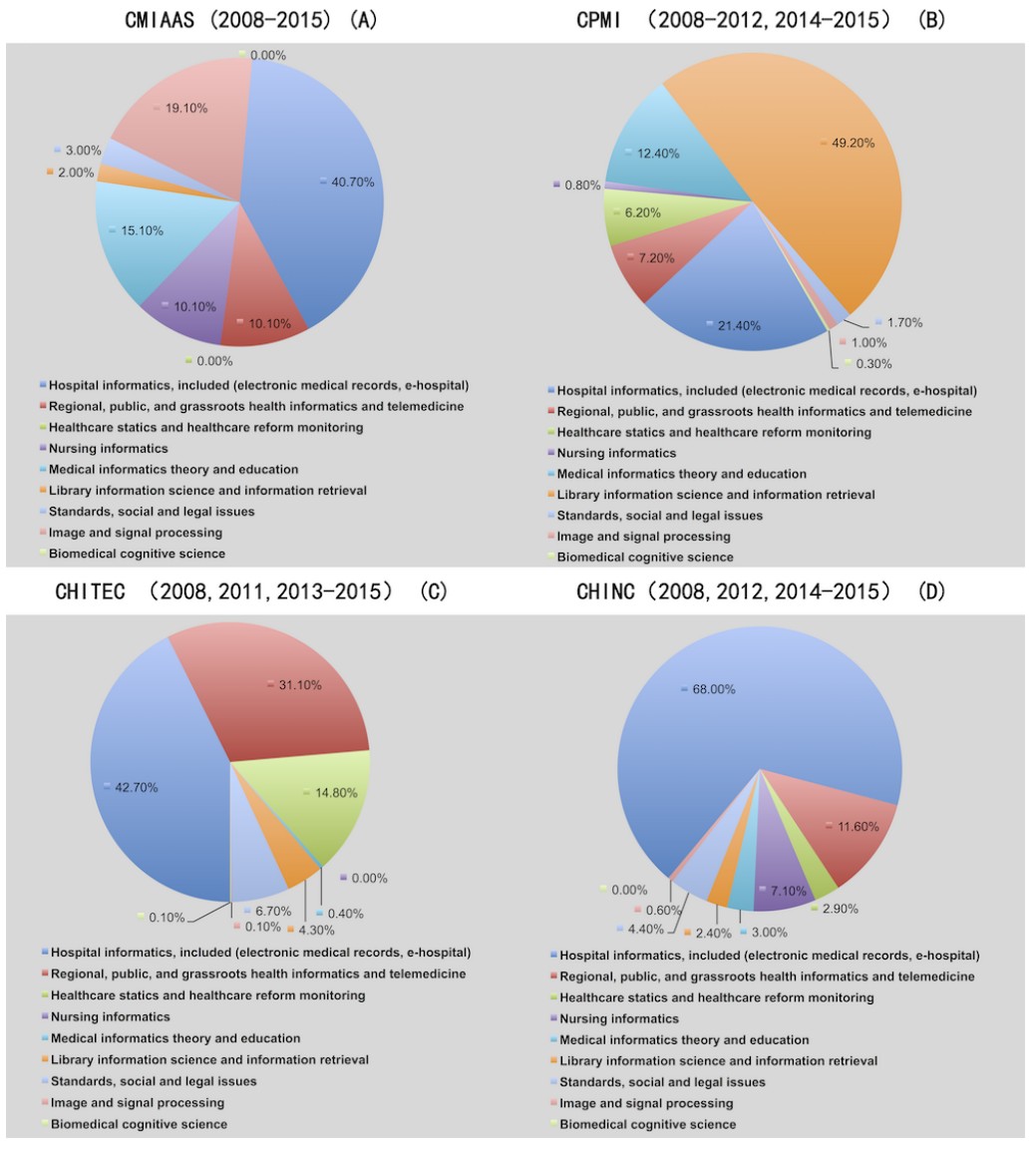

**Figure 5** **Distribution of topics among the CMIAAS, CHINC, CHITEC, and CPMI conference pro-**
**ceedings.** Data show (A) distribution of topics at CMIAAS, (B) distribution of topics at CMPI, (C) distri-
bution of topics at CHITEC, (D) distribution of topics at CHINC.

health care reform in China, which has driven substantial investments in hospital and
population health informatics.

2. Unlike topics in AMIA conference proceedings (*Maojo et al., 2012*), we observed
few topics (as low as <1%) in "medical informatics theory," "biomedical cognitive
science," or "standards, safety, legal, and related issues" in CHINC, CPMI, and CHINC
conference proceedings, and these topics were in AMIA's Call for Papers. Moreover,
many areas in AMIA and HIMSS conference proceedings, such as "dental informatics"
and "consumer health information," were not referenced in the "call for papers" of the

four mainstream Chinese medical informatics conferences. This dearth reflects the gap between the breadth and depth of MI academic research as well as HIT applications in China and the US.

## DISCUSSION

### MI conference proceedings in China: some general observations

1. Weak foundation for MI. In China, the early-stage development of medical informatics was dominated by library science and information, with no support from computer science and information technology (*Lei et al., 2016*); thus, the state of the medical informatics discipline in China is currently behind that of the US and, potentially, other countries.

2. Missing subfields in the medical informatics discipline. Given the internationally recognized categories of subfields in medical informatics, certain subfields remain missing or under-developed in China.

3. The difficulty of using theories to solve practical issues. Few theoretical research studies about medical informatics in China have been carried out, despite the numerous publications in this field. Currently, graduate programs in medical informatics are available in only a few Chinese teaching institutions.

4. Academic institutions in China cannot provide adequate numbers of sufficiently qualified professionals to apply their knowledge of medical informatics in industry (e.g., hospital information technology departments).

Due to the space limitations of this paper, readers may refer to Appendix A-3 (*Unique Characteristics of the Medical Informatics Discipline in China*), if interested in further information.

### MI conferences in China and the US: further analysis and comparisons

For more in-depth investigation of China's current research in medical informatics, the application of HIT, meeting sessions, and the number of participants, subjects discussed at meeting proceedings, academic research, and real-world applications as well as presentations of academic achievements, participating vendors, and review criteria in China's four major medical informatics meetings (i.e., CMIAAS, CHINC, CHITEC, and CPMI, 2015), together with preparation of AMIA (annual symposium and joint summits) and HIMSS (2015) are presented. Figures 6 and 7 throw light on the results, and Table 2 brings forth the specific details.

We analyzed the data in table and figures to reach the main conclusions regarding the status quo of the development of medical informatics in both China and America:

1. According to the cross-field research about the quantitative measures of Sino-US academic and industrial exchange meetings, there is a huge gap between the two countries in terms not only of HIT applications but also of MI academic research.

   1.1. In terms of academic exchange meetings, we compared the relatively large CPMI meeting (2015) with that of the AMIA (including the annual symposium as well as joint summits, 2015) and counted the number of participants as well as meeting

Liang et al. (2017), *PeerJ*, DOI 10.7717/peerj.4082

**Table 2  Summary of the CMIAAS, 2015; CHINC, 2015; CHITEC, 2015; CPMI, 2015; AMIA, 2015; and HIMSS, 2015 conferences.**

| Conference name | Organizer | Inception | Schedule | Meeting duration | Number of attendees | Composition of attendees | Academic achievement | Paper review mechanism | Number of participating companies | Fields covered |
|---|---|---|---|---|---|---|---|---|---|---|
| CMIAAS 2015 | China Medical Informatics Association | 1981 | Every 3 years | 1 day | 200+ | Medical institutions, universities, research institutes | Four forums, 71 conference papers | Format review only | 5 | B, D, I, F, P, T |
| CPMI 2015 | Medical Informatics Branch, Chinese Medical Association | 1993 | Annual | 1 day | 300+ | Medical institutions, research institutes, universities | Four forums, 248 conference papers, including 13 papers presented at the general conference and 48 papers at forums | Format review only | 13 | L, D, B, F, P |
| AMIA 2015 annual symposium | | 1977 | Annual | 4 days | 2,300+ | Medical institutions, universities, research institutions, companies | Ten forums, 14 continuing education classes, and 114 lectures; 156 full-text papers, 80 abstracts, 36 exchange articles, 1,109 posters, 12 system presentations, 7 contests of student-led project design. | Rigorous peer review mechanism | 50 | A, B, C, D, E, F, G, H, I, J, K, L, M, N, O, P, Q, R, S, T |
| AMIA 2015 joint summits | American Medical Informatics Association | 2008 | Annual | 4 days | 1,400+ | | Sixteen forums, 6 continuing education classes, 136 lectures; 66 full-text papers, 158 posters | | 15 | |

**Table 2** (*continued*)

| Conference name | Organizer | Inception | Schedule | Meeting duration | Number of attendees | Composition of attendees | Academic achievement | Paper review mechanism | Number of participating companies | Fields covered |
|---|---|---|---|---|---|---|---|---|---|---|
| CHITEC 2015 | China Institutes of Health Information | 2004 | Annual | 2 days | 3,300+ | Medical institutions, research institutes, government authorities, enterprises, some universities | Ten forums, 87 continuing education lectures; 667 conference papers, of which 37 were nominated as "outstanding papers" | Format review only | 100+ | M, N, K, C, D, I, F, H, Q, T |
| CHINC 2015 | Committee on Information Management, Chinese Hospital Association | 1997 | Annual | 3 days | 3,500 | Medical institutions, enterprises, some universities and research institutes | Seven forums, 99 continuing education lectures; 853 conference papers, of which 71 were nominated as "outstanding papers" | Format review only | 150+ | B, D, I, F, P, T |
| HIMSS 2015 | Healthcare Information and Management Systems Society of the US | 1962 | Annual | 4 days | 38,000+ | Companies, medical institutions, research institutions | More than 250 forums and presentations, more than 300 continuing education lectures, one-day pre-conference seminar | Presenters required to submit abstracts and PPT for peer review | 1,200+, with review performed between 12 regions | L, D, B, F, P |

**Notes.**

A, consumer health informatics; B, clinical information management; C, decision support system; D, electronic medical records; E, medical language processing; F, nursing informatics; G, achievement evaluation; H, public health informatics; I, information retrieval; J, medical cognitive science; K, clinical project management; L, computer-based training; M, coding, classification and terminology; N, clinical guidelines for computerization; O, image, robotics, virtual medical treatment; P, signal processing; Q, standards, social and legal issues; R, dental informatics; S, artificial intelligence; T, telemedicine.
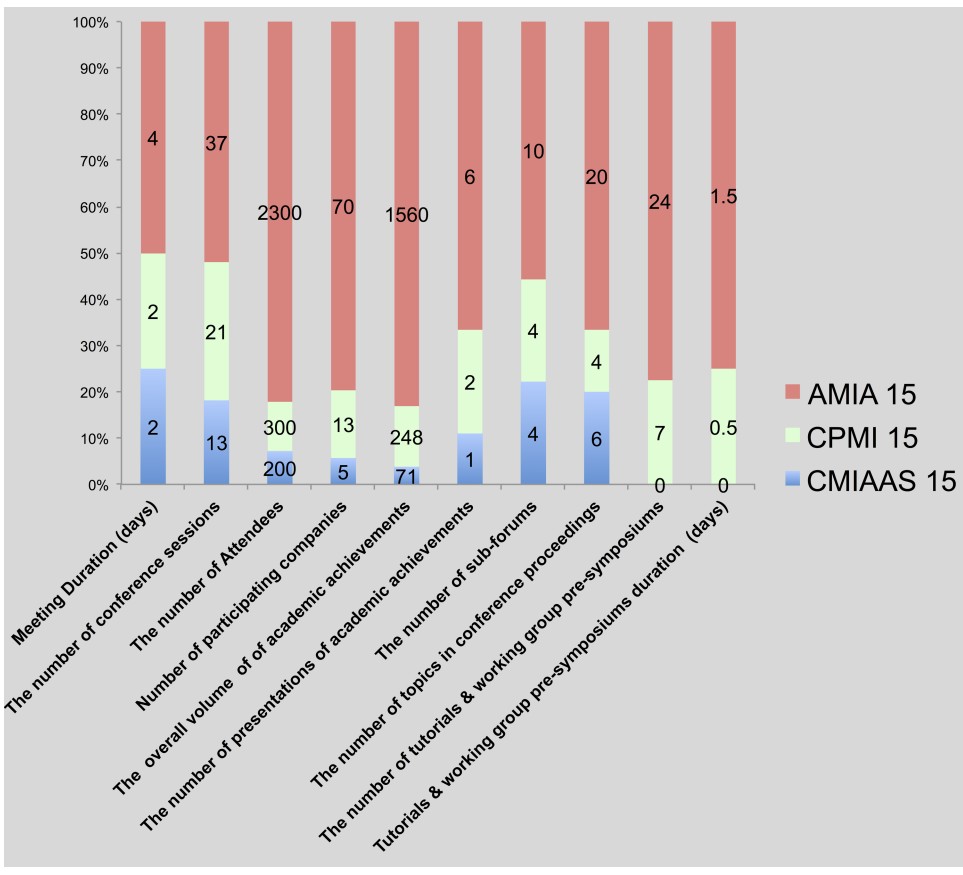

**Figure 6** Comparison of major characteristics of MI conferences (academic) in the US and China, 2015.

papers of the CPMI, which were only approximately 8% of the total number of AMIA meeting papers. For the formats of demonstrating academic achievements, academic achievements were demonstrated at AMIA 2015 in a variety of ways, for instance, system demonstrations, full-text papers, student papers, posters, abstracts, design contests, and descriptions; however, at CPMI 2015, academic achievements were demonstrated in merely two ways: full-text as well as exchange papers. Among these topics, discussions were carried out on merely seven fields at CPMI 2015, and 70.6% of papers were associated with information retrieval as well as hospital informatics; on the contrary, all of the 20 sub-fields associated with medical informatics were transferred at AMIA 2015.

1.2. Regarding HIT application meetings, the gap was huge as well. Our comparison of the relatively large CHITEC 2015 as well as HIMSS 2015 showed that the number of participants at CHITEC 2015 was just 10% of that at HIMSS 2015 (3,500+ compared to 38,000+); the number of participating vendors at CHITEC 2015 was only approximately 8.3% of that at HIMSS 2015 (100+ compared to 1,200+). Academically, as respects value-related measures, CHITEC 2015 included merely
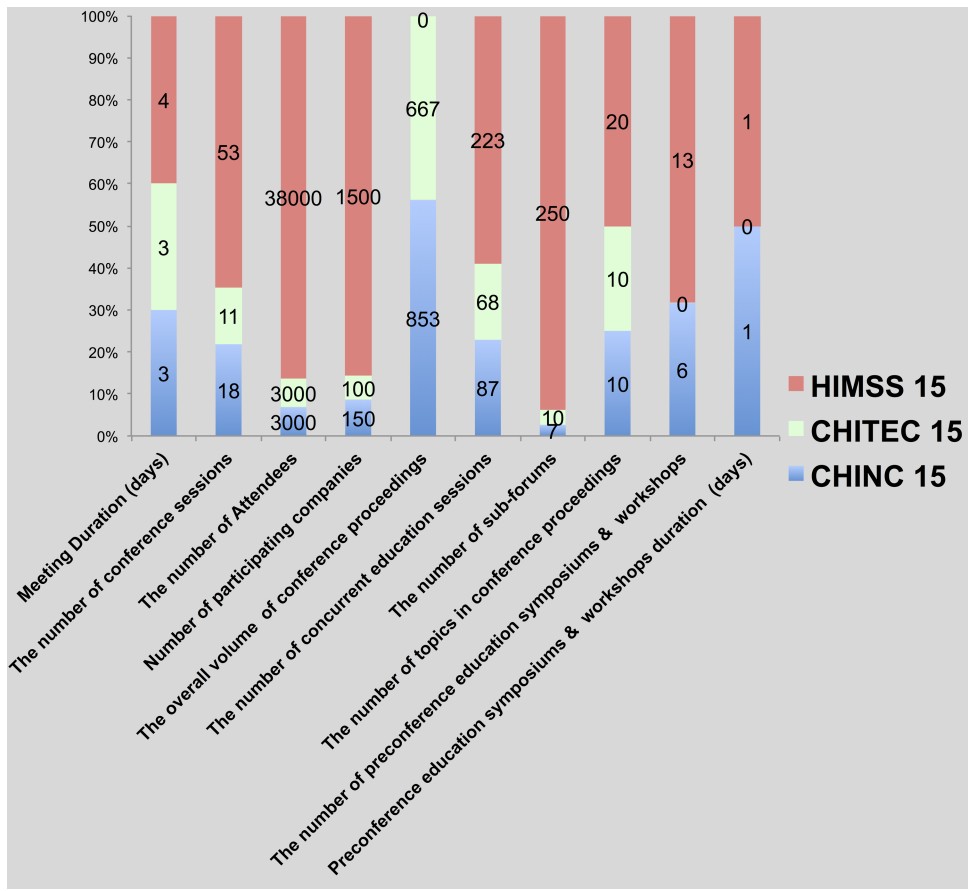

**Figure 7** Comparison of major characteristics of MI conferences (industry) in the US and China, 2015.

10 forums and 87 lectures on continued education, just 4% and 33%, respectively, of the forums and lectures at HIMSS 2015 (250+ forums together with 300+ continuing education lectures over 4 days). In terms of topics, 10 fields were transferred at CHITEC 2015, but 73.8% of meeting papers were associated with population health informatics or hospital informatics; in contrast, all 20 sub-fields associated with medical informatics were discussed at HIMSS 2015.

2. A significant gap between the academic value of Chinese and American MI meeting proceedings papers was also noted. To be more specific, at Chinese HIT meetings, in most cases-only format review was carried out because of insufficient submitted papers and well-rounded peer reviewers who were trained in medical informatics courses. Therefore, the Chinese conference papers' academic value might be relatively low. In contrast, strict peer-review was carried out for AMIA 2015. According to the data available on the website, AMIA's proceedings papers' acceptance rate is lower than 30%; consequently, AMIA is identified by the China Computer Federation Academic Committee as "a significant internationally recognized meeting"(*Committee, 2015*). In addition, in order to further determine the academic value of Sino-American medical

information conferences, we take CPMI '15 and AMIA '15 as an example; we use the Wanfang data retrieval platform and the PUBMED Central database to query paper citations. We found that the 248 CPMI '15 conference papers were cited 15 times, of which only one papers were cited two times, and 13 papers cited one time, with an average citation rate of 0.06. However, the 225 full-text articles from AMIA '15 were cited 67 times, of which one paper was cited four times, one paper cited three times, nine papers cited twice, 42 papers cited once, with an average citation rate of 0.3 times per paper, this is five times that of CPMI' 15.

3. It is a known fact that the big gap in MI conferences between the United States and China is closely associated with their economic strengths and populations as well as their health care systems. 2015 statistics reveal that the United States GDP is 1.78 times that of China (18.04 trillion US dollars (*OECD, 2017a*) vs. 10.14 trillion US dollars (*National Bureau of Statistics of China, 2014a*)), whereas its population is only 23% of the latter (314 million (*OECD, 2017c*) vs. 1.37 billion (*National Bureau of Statistics of China, 2014c*)). However, the United States' expenditure on health care is 16.9% of its GDP (*OECD, 2017b*), while that of China is 5.95% (*National Bureau of Statistics of China, 2014b*). This means that, in the same year, the United States' per capita health expenses amounted to 9451 US dollars (*OECD, 2017b*), whereas Chinese healthcare expenses per capita were only 438 US dollars (*National Bureau of Statistics of China, 2014b*); the former is 21.6 times that of the latter. Moreover, the large gap between the Chinese health system and that of the US suggests that, in the US, new inventions are quickly applied (*Deaton, 2015*). MI, as a new interdisciplinary field with a naturally broad market prospect, has gathered a good amount of investment funding and support, whereby investors would like to use new technologies to reduce medical expenditures, forcing medical institutions to control costs while also guaranteeing quality healthcare. Thus, it is no surprise that the number of participants of HIMSS (2015) is 10 times that of CHITEC (2015), and AMIA (2015)'s academic achievements are 7.8 times those of CPMI (2015). This huge gap implies great room for collaboration between Chinese MI professionals and their US counterparts. For further detailed descriptions, please see Appendix A-3 (*Unique Characteristics of Medical Informatics Discipline in China*).

## CONCLUSION

In China, medical informatics began in the 1970s and was based on library information science. In contrast, in the US, medical informatics is based on computer applications in medicine. At present, a large gap between China and the US exists with respect to research rigor and direction, regional balance in the development of medical informatics disciplines, and the breadth and depth of academic research and industry application. These conclusions are based on our analysis of four Chinese mainstream MI conference proceedings, as well as two major US MI conference proceedings.

In addition, the Chinese medical education curriculum initially focused on "medical literature information systems," which was significantly different from the framework

of the medical education curriculum in the US and other countries. This difference was demonstrated in CPMI proceedings paper topics.

Finally, some relatively large Chinese medical informatics organizations (e.g., CHIMA, SCMI, and CHIA) are not currently IMIA members; CMIA, the only current IMIA member, was previously a secondary institution of the Chinese Institute of Electronics. Consequently, the public status of CMIA, diverging from library information science-based medical informatics, is second only to that of SCMI in the academic community, but compared with CHIMA and CHIA, its business sponsors and attendees are much fewer. Moreover, communication between CHIMA, which focuses on hospital informatics, and CHIA remains a challenge.

This study demonstrates an urgent necessity to elevate the medical informatics discipline in China and to expand research fields in order to maintain pace with the development of medical informatics in the US and other countries. To this end, the authors propose four suggestions: (1) the central Chinese government should establish developmental goals and strategies for the medical informatics discipline; (2) the government should support the integration of medical informatics research with national health informatics; (3) according to international guidelines for medical informatics education prescribed in the latest version of IMIA (*Haux & Murray, 2010*; *Mantas et al., 2010*), a framework based on these guidelines and actual conditions in China should be developed to define the professional qualifications of medical informatics personnel; and (4) CHIMA, CHIA, and SCMI should align their organizational efforts with international MI research fields and extant needs to develop medical informatics in China.

A special note is that, even though conference analysis has offered interesting insights on the current status of MI in China and the US, better measures such as publications and grants will be explored in our next studies. In addition, our current analysis focuses on the development of Medical Informatics in China and in comparison to that in the United States of America. Our next research plan is to include data from Medical Informatics Europe Conferences (MIE) and World Congress on Medical and Health Informatics (MEDINFO), so that the present study can be further extended to the status quo analysis and comparison of China and the European and world medical informatics development.

Because of the space limitations of this paper, readers may refer to Appendix A-4 (*Several Suggestions for the Development of Medical Informatics Discipline with Chinese Characteristics*), if more information is desired.

### Funding

This study was supported by the National Natural Science Foundation of China (NSFC) (grants #81171426 and #81471756) and the Medical and Health Planning Project of Zhejiang Province of China (grant #2017KY386). The funders had no role in study design, data collection and analysis, decision to publish, or preparation of the manuscript.

## Grant Disclosures

The following grant information was disclosed by the authors:
National Natural Science Foundation of China (NSFC): #81171426, #81471756.
Medical and Health Planning Project of Zhejiang Province of China: #2017KY386.

## Competing Interests

The authors declare there are no competing interests.

## Author Contributions

- Jun Liang performed the experiments, analyzed the data, contributed reagents/materials/analysis tools, wrote the paper, prepared figures and/or tables.
- Kunyan Wei analyzed the data.
- Qun Meng and Jiajie Zhang reviewed drafts of the paper.
- Zhenying Chen performed the experiments, contributed reagents/materials/analysis tools.
- Jianbo Lei conceived and designed the experiments, analyzed the data, wrote the paper, prepared figures and/or tables.

## Data Availability

The raw data is included in the tables and figures in the article.

## Supplemental Information

Supplemental information for this article can be found online at http://dx.doi.org/10.7717/peerj.4082#supplemental-information.

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
