# Peer review of "Development of medical informatics in China over the past 30 years from a conference perspective and a Sino-American comparison"

_PeerJ, doi:10.7717/peerj.4082_

## Round 0.1 · original submission · Major Revisions

· Academic Editor

Major Revisions

Three reviewers suggested revision of the manuscript (between major and minor revision). Please take care on figures and legends. Each figure should show some result discussed in the text, not just statistics.
I suggest you change the paper title to make it more common - remove word 'for 2015' in the title. '30 years perspective' phrase is already self-explanatory. Perspectives of medical informatics development review could be more interesting for readers in world-wide perspective, including publications from Europe, India. It is advice for future. Hope you revise this manuscript soon.

Thank you again for discussion of interesting problem of medical informatics development.

·

Basic reporting

This paper is well structured and written.

Experimental design

The data used in this paper cover 30 years. The results are comprehensive and constructive.

Validity of the findings

Review the progress of medical informatics in China over the past 30 years to summarise the gains and losses even give constructive suggestions for future development is an important task for the whole country.

Additional comments

To show the development and do a comparison from a conference perspective, especially academic conferences, it would be better to show the impact of the precedings. An impact can be the number of citations or downloads of the publications of a conference.

Reviewer 2 ·

Basic reporting

The English language in the manuscript should be improved. Others are fine.

Experimental design

No comment

Validity of the findings

Data is statistically sound and controlled. Conclusion are well stated.

Additional comments

In this manuscript, the authors compared four main Midical Informatics (MI) conferences held in China, and two main MI conferences held in USA from different perspectives. Some conferences are more industry-oriented, and some are academy-oriented. The authors concluded that, the conferences reflected that the Midical Informatics discpline in China is behind of that in USA, either in participants or topics involved. The content of the manuscript is informative, while the English language needs to be improved.

The followings are some specific points in the manuscript.

1) Something is wrong in the lines 88-90. The market was "23.7" billion US dollars. After 5 years, it will shrink to "14.5" billion US dollars.
In 2015, the market of hospital informatics alone reached 23.7 billion US dollars; by 2020, this market is projected to exceed 14.5 billion US dollars, at an annual compound growth rate of ≥ 24%


2) line 160: "choose" should be in the past tense "chose". Similar for many other verbs

3) lines 271-275: It is better to list "health administrative authorities" as a specific category for the comparison purpose. If it is treated as "other", it is difficult for the readers to identify the difference from the Figures.

4) line 276: what's the meaning of SCMI?

5) The last Figure is Figure 7, while the figure legend is Figure 6(b). Need to be corrected.

Reviewer 3 ·

Basic reporting

no comment

Experimental design

no comment

Validity of the findings

The 2020 estimate of the market volume (p.6) seems to be either miscalculated or mistyped.
I suggest to be more specific in Results and Conclusions of the Abstract.
Most of the figers and half of the tables don't touch USA MI therefore I suggest to exclude "the Sino-American comparison..." from the paper's title and to compare Chinese conferences in more detail.
P.12: The high variability in the paper's quality is intuitively acceptable but not proved. The low acception value may indicate the funding bias as well. Therefore I suggest to decrease the level of confidence when stating the conclusions about this.

Additional comments

I liked the paper: the authors managed to address an important issue by simple and transparent methods. The paper is written in excellent English and I've read the Appendix with special interest.

---

## Round 0.2 · accepted · Accept

· Academic Editor

Accept

Dear authors, two reviewers recommend accept the paper after revision. The third reviewer was invited, but did not reply. I confirm that the article has been revised sufficiently to be accepted now.

·

Basic reporting

no comments

Experimental design

no comments

Validity of the findings

no comments

Additional comments

In this revised version, the authors add a new part about preceding impact. The new finding from impact perspective is consistent with the conclusion of the whole paper. I feel this work meets the criteria as a research publication and suggest to accept this paper.

Reviewer 3 ·

Basic reporting

no comment

Experimental design

no comment

Validity of the findings

no comment